

# The role of GPT in promoting inclusive higher education for people with various learning disabilities: a review

Thippa Reddy Gadekallu[1,2,3,4,5,*], Gokul Yenduri[6,*], Rajesh Kaluri[2], Dharmendra Singh Rajput[2], Kuruva Lakshmanna[2], Kai Fang[7], Junxin Chen[8] and Wei Wang[9,10]

[1] Department of Electrical and Computer Engineering, Lebanese American University, Byblos, Lebanon
[2] School of Computer Science Engineering and Information Systems, Vellore Institute of Technology University, Tamil Nadu, India
[3] Zhongda Group, Jiaxing, China
[4] College of Information Science and Engineering, Jiaxing University, Jiaxing, China
[5] Division of Research and Development, Lovely Professional University, Phagwara, India
[6] School of Computer Science and Engineering, VIT-AP University, Amaravati, Andhra Pradesh, India
[7] School of Mathematics and Computer Science, Zhejiang Agricultural and Forestry University, Hangzhou, China
[8] School of Software, Dalian University of Technology, Dalian, China
[9] Guangdong-Hong Kong-Macao Joint Laboratory for Emotional Intelligence and Pervasive Computing, Artificial Intelligence Research Institute, Shenzhen MSU-BIT University, Shenzhen, China
[10] School of Medical Technology, Beijing Institute of Technology, Beijing, China
[*] These authors contributed equally to this work.

Corresponding author
Wei Wang, ehomewang@ieee.org

## ABSTRACT

The generative pre-trained transformer (GPT) is a notable breakthrough in the field of artificial intelligence, as it empowers machines to effectively comprehend and engage in interactions with humans. The GPT exhibits the capacity to enhance inclusivity and accessibility for students with learning disabilities in the context of higher education, hence potentially facilitating substantial advancements in the field. GPT can provide personalized and diverse solutions that successfully cater to the distinct requirements of students with learning disabilities. This motivated us to conduct an extensive review to assess the effectiveness of GPT in enhancing accessibility and inclusivity in higher education for students with learning disabilities. This review offers a comprehensive analysis of the GPT and its significance for enhancing inclusivity in the field of higher education. In this research, we also examined the possible challenges and constraints associated with the integration of GPT into inclusive higher education, along with potential solutions. Overall, this review is intended for educators, students with and without learning disabilities, policymakers, higher education institutes, researchers, and educational technology developers. This review aims to provide a comprehensive understanding of GPT in promoting inclusive higher education for people with various learning disabilities, its impacts on inclusive higher education, emerging challenges, and potential solutions.

# INTRODUCTION

During the course of educational development, the effort to improve access to higher education for students with a variety of learning disabilities has faced numerous challenges, and also yielded notable successes (*Subotnik, Olszewski-Kubilius & Worrell, 2011*). In many countries around the world, the field of higher education has seen an unprecedented rate of growth during the last six decades (*Stephens et al., 2008*). According to the United Nations Educational, Scientific and Cultural Organization (UNESCO), there has been significant growth in the average enrolment rate in higher education, rising from less than 10% in 1970 to 41% in 2017 (*Brunner & Labraña, 2020*; *Salmi, 2023*). Many developing countries have achieved notable gains in expanding access to higher education for a larger population. Students with learning disabilities may have academic challenges, even if they are given the chance to register and participate, unless they get adequate assistance (*Cai & Richdale, 2016*; *Boztaş et al., 2023*). The principal aim of the expanding domain of inclusive education is to provide equitable educational opportunities for students across a spectrum of cognitive and physical disabilities (*Opertti & Belalcázar, 2008*). The field of higher education is undergoing transformation in light of the integration of advanced technologies, such as artificial intelligence (AI), extended reality (XR), blockchain, and the metaverse (*Kaddoura & Al Husseiny, 2023*). The use of AI significantly improved the accessibility of technological advancements, especially within the field of education (*Grassini, 2023*). The use of AI in assistive technology has accelerated advancement and enhanced the skills of students with learning difficulties (*Alam, 2021*). Furthermore, the progress made in the field of machine learning (ML) has resulted in the emergence of novel technologies that provide extensive prospects for assisting and facilitating students with learning difficulties in their day-to-day activities (*Thieme, Belgrave & Doherty, 2020*). AI has empowered students with learning disabilities to participate in academic activities on an equal footing with their peers. Students who have visual, auditory, and cognitive difficulties are currently experiencing advantages from technological breakthroughs such as speech-to-text transcription and dynamic captioning, among others (*Fossey et al., 2017*).

The development of modern technology aimed at addressing complex requirements eventually produces benefits for everyone. Numerous technologies, that were originally designed for use by those with disabilities, have subsequently gained widespread adoption among the broader populace. AI advancements, like chatbots, image recognition, autonomous cars, and speech-to-text transformation are advantageous for both people and society at large (*Martínez-Plumed, Gómez & Hernández-Orallo, 2021*). One such recent development in AI is the generative pre-trained transformer (GPT), which has enormous potential to support inclusive higher education for individuals with a range of learning disabilities (*Rasul et al., 2023*). A more conversational and natural method of engaging with computers is provided by the GPT. Open AI has created a family of natural language models of GPT. This paradigm is sometimes called generative AI because of its ability to produce unique outcomes. To provide users with textual replies based on AI, GPT makes use of natural language processing and learns from large amounts of data. This makes interactions with the system seem more conversational and real (*Zhang & Li, 2021*).

Considering the aforementioned possible benefits of GPT, the model can also be advantageous for inclusive higher education, including in improving learning experiences and accessibility for individuals with various learning difficulties. This served as a reason for us to undertake this review.

## Paper organization

The structure of this paper's organization is illustrated in Fig. 1. "Introduction" sets the scene for our study on how GPT technology helps in making higher education more inclusive for people with different learning disabilities. In "Survey methodology", we reviewed existing research in this area, giving a detailed overview of what has already been studied. Then, in "Preliminaries", we explained the basics of GPT technology, what learning disabilities are, and the idea of inclusive higher education. This part was important to show why we conducted our study and what the current situation is. "Impact of GPT in inclusive higher education" focused on how GPT can make higher education more accessible to people with learning challenges, by linking GPT's tech capabilities to the specific educational needs of these people. In "Projects", we shifted our focus to how GPT is used in real life, by discussing different research projects and industry efforts that use GPT. This showed how our theoretical ideas are applied practically. In "Challenges and potential solutions" we dealt with the challenges of using GPT in teaching methods that include everyone, and discussed possible solutions, taking into account both the technical and ethical sides. Finally, in "Conclusion and future directions", we concluded our paper by summarizing the main findings and the contributions of our research. We also thought about the overall implications of our results and suggested ideas for future research in this field. The list of key acronyms is listed in Table 1.

## Related works and contributions

According to a recent analysis by Bloomberg Intelligence (BI), the generative AI industry is expected to develop rapidly, reaching $1.3 trillion over the next 10 years from a market size of around $40 billion in 2022, because of the entry of consumer generative AI projects like OpenAI's ChatGPT. According to BI's study, growth may accelerate with training infrastructure driving the market in the short term and progressively moving to inference devices for large language models (LLMs), digital advertisements, specialized software, and services in the medium to long term (*Bloomberg, 2023*). The capabilities of GPT started showing results in a diverse array of possible applications across several domains, such as education, healthcare, tourism, gaming, construction and design, and e-commerce (*Bommasani et al., 2021*; *Sahib et al., 2023*).

The maturity and advancement of natural language models led by GPT are generating great potential for healthcare applications. The review by *Zhang et al. (2023d)* summarized the possible application prospects of GPT in healthcare, assessed the limitations that still exist, and suggested possible improvements. They concluded that the current version of GPT can provide accurate and practical information for doctors and patients, help healthcare professionals diagnose and treat, promote medical education, improve the accuracy and efficiency of the healthcare system, etc. They also highlighted some limitations,

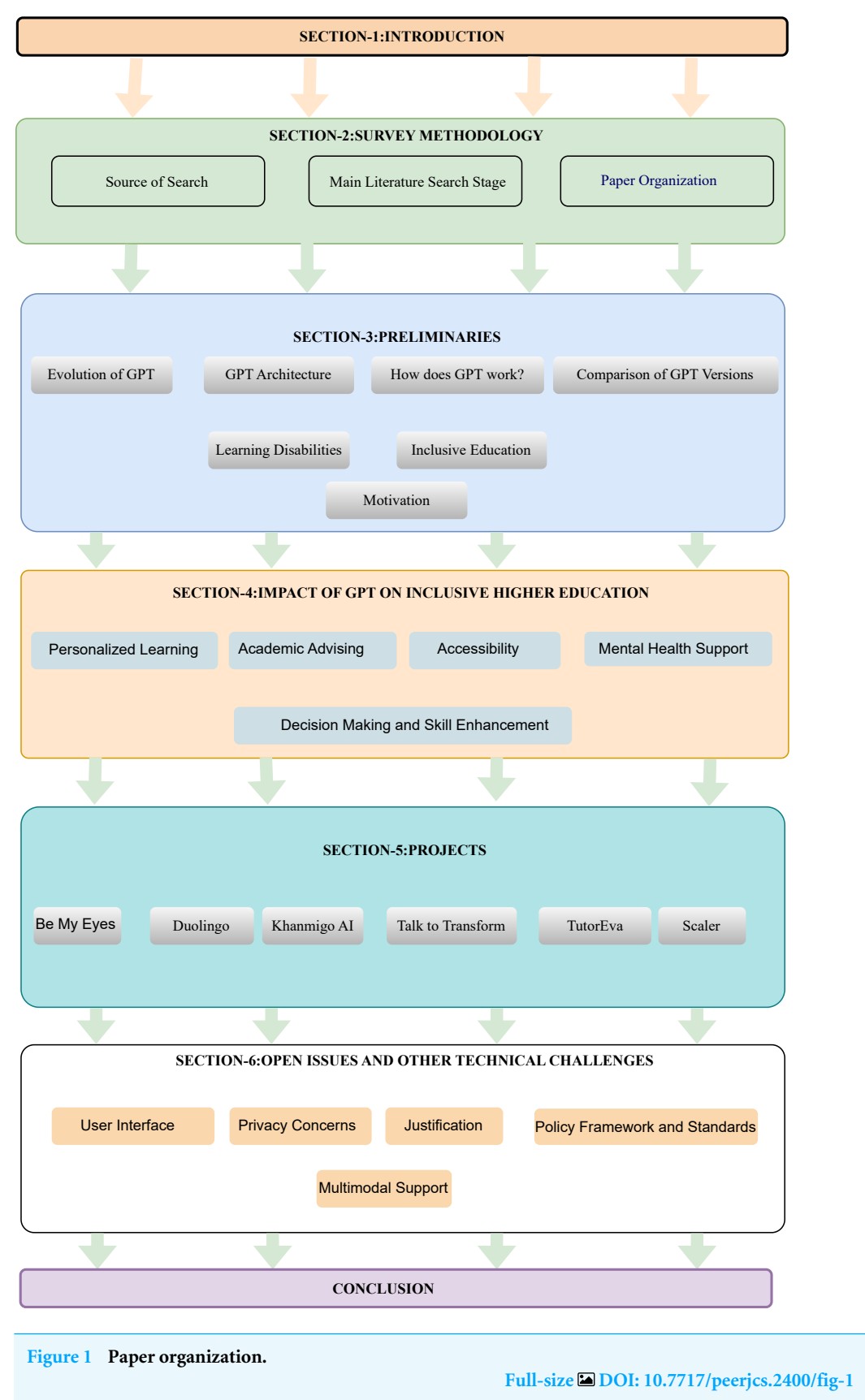

**Figure 1   Paper organization.**

**Table 1  List of key acronyms.**

| Acronyms | Description |
| --- | --- |
| ADHD | Attention-Deficit/Hyperactivity Disorder |
| AI | Artificial Intelligence |
| BCI | Brain Computer Interface |
| BI | Bloomberg Intelligence |
| CAPD | Central Auditory Processing Disorder |
| DCD | Developmental Co-ordination Disorder |
| GPT | Generative Pre-trained Transformer |
| GAI | Generative Artificial Intelligence |
| HCI | Human-Computer Interaction |
| LLM | Large Language Model |
| MHD | Mental Health Disorders |
| ML | Machine Learning |
| NDD | Neuro Developmental Disorder |
| NVLD | Nonverbal Learning Disorder |
| NGSS | Next Generation Science Standards |
| PD | Psychological Disorders |
| UNESCO | United Nations Educational, Scientific and Cultural Organization |
| UI | User Interface |
| XAI | Explainable AI |
| XR | Extended Reality |

such as the accuracy of the information, the privacy of the data, and ethical and moral sentiments, that still need to be addressed. They also suggested improving accurate and effective medical applications with GPT and highlighted the need to promote the standardization of ethical and copyright issues (*Zhang et al., 2023d*).

*Yenduri et al. (2023b)* conducted a review of the widespread use and effectiveness of GPT. According to the authors, GPT has attracted a great deal of interest from the research and industry sectors due to its extraordinary performance in a variety of natural language processing tasks, and its capacity to interact like humans. They presented a comprehensive overview of the GPT, including its design, operational procedures, training methodologies, technologies, and effects on a variety of applications. In this evaluation, the potential limitations and constraints of GPT were also investigated. In addition, they discussed potential future directions and solutions. The primary objective of their research is to gain an in-depth understanding of GPT and its associated technologies, as well as their impact on diverse applications such as education, healthcare, industry, agriculture, travel and transport, e-commerce, entertainment, lifestyle, gaming, marketing and finance, and to explore possible solutions to address these challenges (*Yenduri et al., 2023b*).

The adoption of GPT is anticipated to result in significant changes within the hotel and tourism industries, as it will disrupt the processes by which consumers request information and make decisions, as well as how organizations generate, develop, and provide personalized amenities and experiences. In their study, *Gursoy, Li & Song (2023)*

conducted a comprehensive analysis of the benefits provided by GPT, as well as the potential challenges and risks they present to the hotel and tourism industries. In addition, the discussion assessed the viability of incorporating GPT into various phases of travel, and the decision-making processes associated with these phases. The researchers concluded their study by proposing future research work centered on the use of GPT technology for the development and provision of hospitality and tourism experiences. This proposed research has the potential to provide useful direction for future enhancements (*Gursoy, Li & Song, 2023*).

Quests serve a crucial role in the framework of role-playing games (RPGs). The prevailing practice in the gaming industry is the manual creation of narrative-driven quests, despite the increasing demand from users for more extensive and immersive game content, as well as the commercial need for sustained player engagement. However, to meet these expectations and needs, alternative approaches that use procedural quest generation methods are becoming necessary. Nevertheless, recent advancements in AI have resulted in the creation of generative language models that exhibit potential computational storytelling capabilities. In contrast, traditional methods tend to provide quest descriptions that are mostly repetitive and lack creativity. In their study, *Värtinen, Hämäläinen & Guckelsberger (2022)* use the two widely adopted transformer models, namely GPT-2 and GPT-3, to procedurally generate quest descriptions in RPG video games. A comprehensive dataset including 978 missions and their corresponding descriptions from six distinct role-playing games was gathered, subjected to rigorous analysis, and then released for public access. The researchers implemented a series of optimizations based on many mini-studies to improve the performance of GPT-2 using the given dataset. The final Quest-GPT-2 model was verified through an online user survey administered to a sample of 349 individuals engaged in role-playing activities. According to the data, it was determined that just 20% of quest descriptions are approved by a human reviewer. They also provide recommendations on applications for Quest-GPT-2. This is complemented by case studies on GPT-3 to highlight the future potential of state-of-the-art natural language models for quest generation (*Värtinen, Hämäläinen & Guckelsberger, 2022*).

The construction sector could transform by automating time-consuming and repetitive processes through the use of GPT models like ChatGPT. A construction schedule for a basic construction project was created using ChatGPT in the research presented by *Prieto, Mengiste & García de Soto (2023)*. Participants in the pool assessed ChatGPT's output by giving their opinions on the system's overall quality and interaction experience. To meet the criteria of the specified scope, the findings demonstrate that ChatGPT can provide a logical timetable that makes sense. According to the participants, the technology has the ability to automate several preparatory and time-consuming procedures, and their overall experience was favorable. The authors stated that the GPT models still have to be developed further before the technology can be extensively used in the industry. Overall, their study emphasizes the benefits of GPT models for the construction sector, as well as the need for more research (*Prieto, Mengiste & García de Soto, 2023*).

LLMs that follow instructions, such as ChatGPT, have recently demonstrated extraordinary performance in a variety of natural language processing (NLP) tasks.

However, generic LLMs face significant obstacles due to the unique characteristics of e-commerce data. It is crucial to have an LLM designed for e-commerce situations with robust cross-dataset and task generalization capabilities. To address this issue, *Li et al. (2023)* created the first e-commerce instruction dataset, EcomInstruct, with a total of 2.5 million instruction data. By constructing atomic tasks with fundamental E-commerce data types, such as product information and customer evaluations, EcomInstruct expands both the volume and the variety of data. Atomic tasks are intermediate tasks that are implicitly engaged in resolving an ultimate task, which they refer to as chain-of-task activities. They used EcomInstruct to train the backbone model BLOOMZ, from which they constructed EcomGPT with varying parameter scaling. Using the fundamental semantic comprehension skills learned from the chain-of-task assignments, EcomGPT demonstrated exceptional zero-shot generalization capabilities, according to their findings. Comprehensive experiments and human evaluations demonstrate that EcomGPT outperforms ChatGPT when it comes to the generalization of e-commerce tasks across datasets and tasks (*Li et al., 2023*).

In higher education, the implementation of AI can profoundly alter the ways in which we teach and learn. GPT is one such tool that has the potential to offer students personalized recommendations, promote increased collaboration and communication, and improve student learning outcomes. However, obstacles remain, including challenges related to implementation and ethical considerations. Previous studies on the application of AI in education were analyzed by *Rawas (2023)*, with particular emphasis on ChatGPT and its potential applications in higher education. In addition to implementation recommendations, the study evaluated the benefits and drawbacks of utilizing ChatGPT in higher education. The study also examined the future opportunities for future ChatGPT research in the field of higher education. The findings of this research suggest that ChatGPT presents a significant opportunity for institutions of higher education to improve the overall quality and availability of education. Yet, its implementation requires a cautious approach and a comprehensive understanding of the potential benefits and obstacles involved (*Rawas, 2023*).

In another study, *Strzelecki (2023)* has proposed a model that draws upon a previous theory of technology adoption. In order to construct a predictive model capable of accurately discerning the actions and utilization patterns of ChatGPT users, a selection of seven parameters was made. The author employed the partial least squares approach of structural equation modeling to analyze the dataset. The model demonstrated a high level of reliability and accuracy. The findings of this study were derived from self-reported data, collected from a sample of 534 students enrolled at a state institution in Poland. The results confirmed the majority of the first proposed solutions, specifically nine out of ten. The findings of the study indicated that habit emerged as the most prominent determinant in predicting behavioral intention, with performance expectancy and hedonic incentive ranking subsequently. The study revealed that behavioral intention emerged as the key determinant of use behavior, followed by personal innovativeness as a subsequent predictor. The study emphasized the need for more research into the potential implementation of AI tools in educational practices (*Strzelecki, 2023*).

In their study, *Currie et al. (2023)* evaluated ChatGPT's performance on various written tasks among a sample of six undergraduate students enrolled in medical radiation science courses. Their findings from the study indicated that ChatGPT's performance fell below the average level across all assessed tasks. However, it exhibited superior performance in examinations pertaining to foundations or general subjects. Their work posits that ChatGPT presents a potential threat to academic integrity, while simultaneously providing opportunities for improved learning settings. The limits of this system encompass its constrained ability to facilitate student dishonesty, the possibility of introducing inaccuracies, and the potential for the dissemination of AI-generated content (*Currie et al., 2023*).

The aforementioned recent studies, as shown in Table 2, demonstrate the notable influence of GPT and its importance in several industries. Additionally, there is an abundance of reviews and research works available relating to several fields. There is currently a lack of research and academic literature regarding the use of GPT models in the advancement of inclusive higher education for individuals with various learning difficulties. Therefore, the above observations have motivated us to conduct this in-depth review of the role of GPT in promoting inclusive higher education for people with various learning disabilities.

## SURVEY METHODOLOGY

In order to meet the objectives of this review, which is intended for educators, students with and without learning disabilities, policymakers, higher education institutes, researchers, and educational technology developers, the process of conducting a literature search was divided into two distinct sections.

### Source of search

This systematic literature review provides a comprehensive overview of the role of GPT in promoting inclusive higher education for people with a range of learning disabilities, based on rigorous analysis of scholarly literature from multiple credible sources. Our research predominantly concentrated on scholarly publications that endure a peer-review process, as well as reputable national and international conferences, websites, seminars, books, symposiums, and journals known for their high-quality content. To establish the credibility and legitimacy of our sources, we examined reputable archives such as Google Scholar and arXiv, as well as publications from renowned databases such as IEEE, Taylor & Francis, Elsevier, Springer, and Wiley, among others.

We used specific keywords, including

- GPT-powered accessibility tools
- GPT for inclusive education
- GPT-driven learning support
- Assistive AI technologies
- Inclusive online learning with GPT
- AI-powered accommodations

- Personalized learning with GPT
- Inclusive curriculum enhancement using GPT,

to identify relevant sources and publications of GPT in promoting inclusive higher education for people with various learning disabilities. The limitations and potential biases of the selection criteria in this review may exclude valuable insights from less formal sources, or emerging research that has not yet been published in established journals, as well as factors like keyword selection and language.

## Main literature search stage

In this stage, a comprehensive examination of all the collected articles was conducted, with the screening procedure including a review of the titles and the exclusion of papers with substandard content. Examining the abstracts of the remaining publications to determine their respective contributions was the next step. During the final phase of our literature review, we carefully retrieved the necessary data for our analysis. By following these steps, we ensured that our research was based on credible, high-quality sources (*Kitchenham et al., 2009*).

### Inclusion criteria

In order to qualify for inclusion in this review, the literature must concentrate predominantly on methodologies which include GPT and aim to promote inclusive higher education for students with a wide range of learning disabilities. Furthermore, we included research that employs innovative pedagogical strategies or specialised assistive technologies to support students with learning disabilities. Furthermore, we inculcated conference and journal articles that analyse the outcomes and optimal methodologies of various inclusive educational models, including those that leverage GPT, in order to enhance the learning experience of individuals with learning disabilities.

### Exclusion criteria

Any literature that does not specifically address strategies for encouraging inclusive higher education for students with a range of learning disabilities will not be included in this study. Moreover, works that do not give priority to GPT or creative instructional approaches for students with learning disabilities will be excluded. Articles that do not fit into the conference or journal publication category, or that do not provide a sufficient analysis and comparison of the impacts of different inclusive educational models related to GPT, will also be excluded. In summary, the inclusion criteria focus on GPT in higher education for students with learning disabilities to understand GPT's unique role in this field. We include studies on innovative teaching strategies and technologies to capture the evolving nature of inclusive education. The exclusion criteria remove studies not centered on inclusive education or GPT, ensuring the review remains focused and relevant. We also exclude non-academic sources to maintain the research's credibility and depth. These criteria ensure a comprehensive and targeted analysis of GPT's impact in inclusive educational environments.

**Table 2  Summary of Related works on GPT.**

| Ref | Contributions | Remarks |
|---|---|---|
| *Zhang et al. (2023d)* | This study investigated the summarized the possible application prospects of GPT in **healthcare**. | This study did not provide information on how GPT promotes inclusive higher education for people with learning disabilities in the healthcare sector. |
| *Yenduri et al. (2023b)* | This study conducted a review of the widespread use and effectiveness of GPT in **various sectors including education**. | This study did not provide insights on how GPT as an assistive technology can help on promote inclusive higher education for people with learning disabilities. |
| *Gursoy, Li & Song (2023)* | This study conducted a comprehensive analysis of the benefits provided by GPT as well as the potential challenges and potential risks it presents to the **hotel and tourism industries**. | This research did not address the use of GPT as an assistive technology for students with learning impairments in the context of inclusive higher education within the hotel and tourist sectors. |
| *Värtinen, Hämäläinen & Guckelsberger (2022)* | This study compared GPT-2 and GPT-3 purpose of procedurally generating quest descriptions in **RPG video games**. | The present research did not specifically investigate the challenges associated with the implementation of GPT-2 and GPT-3 models for students with learning disabilites within the context of educational gaming. |
| *Prieto, Mengiste & García de Soto (2023)* | This study examined the GPT in the aspect of reduction of time-consuming and repetitive processes in the **construction sector**. | This study did not specifically address the unique challenges faced by students with learning disabilities in the context of inclusive higher education in civil engineering when integrating GPT as an assistive technology. |
| *Li et al. (2023)* | This work highlights the need of LLM designed for **E-commerce** situations with robust cross-dataset/task generalisation capabilities. | The works also highlights the need of domain-specific LLM designs. |
| *Rawas (2023)* | This work highlighted the benefits and drawbacks of GPT in **higher education**. | The works did not provide any benefits or highlight the need for GPT in inclusive higher education. |
| *Strzelecki (2023)* | This work analyzed the students' acceptance and use of GPT in **higher education**. | The analysis conducted in this study focuses primarily on the general student population, without taking into account the inclusion of students with learning disabilities in higher education. |
| *Currie et al. (2023)* | This work analyzed the GPT performance in **medical courses** and highlighted the challenges. | The primary focus of this study pertains to the performance of GPT in the context of higher education. However, it does not address the potential of GPT to facilitate inclusive higher education or adequately emphasize the obstacles associated with this perspective. |
| Our work | This work provides an in-depth review of GPT in promoting inclusive higher education for people with various learning disabilities | |

## PRELIMINARIES

The following section provides a detailed overview of GPT, learning disabilities, inclusive higher education, and the motivation for the work.

## GPT

GPTs are a type of LLM and a renowned framework for generative artificial intelligence (*Meskó & Topol, 2023*). Even though the concept of generative pretraining (GP) in machine learning applications was well-established, the transformer architecture was not created until 2017 (*Chan, 2023a*). In 2018, LLMs such as BERT, a pre-trained transformer (PT) that was intended to be an "encoder-only" model, rather than generative, emerged as a result of these advancements (*Devlin et al., 2018*). The first GPT was unveiled by OpenAI in 2018 around the same time (*Finnie-Ansley et al., 2022*).

GPT is a deep learning model that can be customized for applications including text categorization, sentiment analysis, machine translation, and language modeling (*Minaee et al., 2021*; *Huang et al., 2021*). It has been pre-trained on enormous amounts of textual data. The design of the GPTs is a significant advance over prior NLP techniques (*Han et al., 2021*). The implementation of a self-attention mechanism improves the model's language comprehension and production, because it can consider the context of the entire phrase when producing the next word. The decoder generates the output text based on the input text (*Chavez et al., 2023*).

LLMs have significantly advanced as a consequence of the recent advancements in transformer-based language models, and their use in a variety of applications is growing by using them as pre-trained models, and fine-tuning them for certain tasks (*Min et al., 2023*; *Ali et al., 2023*). Researchers have shown that these models capture not just linguistic information but also the generic knowledge underlying the data (*Schramowski et al., 2022*). They have thus effectively improved the state of the art for several NLP tasks, suggesting that they have the potential to further transform AI.

From GPT-1 to the most current GPT-4, there have been notable improvements in the design, capabilities, and scalability of GPT models, as shown in Table 3 (*Wu et al., 2023*). GPT-1, which was released in June 2018, was trained using a large 4.5 GB text dataset sourced from BooksCorpus from 7,000 unpublished novels across many genres. Its transformer decoder design consists of 12 levels and 12 heads, and it includes 117 million parameters (*Nassiri & Akhloufi, 2023*). Released in February 2019, GPT-2 has grown to include 1.5 billion parameters, which enables it to learn from 40 GB of web information from 8 million pages and generate logical and contextually rich text (*Zhang et al., 2023a*).

GPT-3 went beyond this trend. It is trained on a dataset of 499 billion tokens from many sources, including CommonCrawl, WebText, the English Wikipedia, and book corpora, with 175 billion parameters (*Dale, 2021*; *Chen, 2022*). Its introduction in May 2020 marked a breakthrough in the production and interpretation of natural language.

Even though GPT-3.5 and GPT-4 are still pushing the boundaries of AI, the details of their architectures and training sets are still undisclosed. AI has progressed further with the release of GPT-4 in March 2023, since it is now capable of processing images in addition to text. GPT-4's estimated training cost of 2.1e25 FLOP emphasizes the incredible processing resources required to power these state-of-the-art AI advances. These models' expanding size and complexity come at a significant computational expense (*Thirunavukarasu et al., 2023*). The general transformer architecture of GPT is depicted in Fig. 2 and the transformer

**Table 3  Various versions of GPTs.**

| Model | Architecture | Parameters | Training data | Year of release |
|-------|-------------|------------|---------------|-----------------|
| GPT-1 | 12-level, 12-headed Transformer decoder | 117 million | BookCorpus with 4.5 GB of text, from 7,000 unpublished books | 2018 |
| GPT-2 | GPT-1 modified with normalization | 1.5 billion | WebText: 40 GB of text, 8 million documents, from 45 million webpages | 2019 |
| GPT-3 | GPT-2 with allowed larger scaling | 175 billion | 499 Billion tokens consisting of CommonCraw, WebText, English Wikipedia, and books corpora | 2020 |
| GPT-3.5 | Undisclosed | 175 billion | Undisclosed | 2022 |
| GPT-4 | Also trained with both text prediction and RLHF; accepts both text and images as input | Undisclosed | Undisclosed | 2023 |

architecture along with input transformations for fine-tuning on different tasks is depicted in Fig. 3 (*Yenduri et al., 2023b*).

## Learning disabilities

Learning disabilities involve a diverse array of neurological conditions that have a significant impact on a student's capacity to acquire knowledge, comprehend, and process information, as well as successfully communicate within conventional educational environments (*Siegel & Heaven, 2013*). Dyslexia is known to impede the processing of language, while dyspraxia is recognized for its impact on movement abilities (*Hendrickx, 2010*). Dyscalculia has a significant influence on an individual's ability to comprehend mathematical concepts, whereas dysgraphia adversely affects their writing skills (*Döhla & Heim, 2016*; *Vigna et al., 2022*). Auditory processing disorder (APD) is characterized by disruptions in the perception of auditory stimuli, whereas visual processing disorder (VPD) is associated with impairments in the processing of visual information (*Sardone et al., 2019*). Nonverbal learning disorder (NVLD) is characterized by impairments in spatial and organizational abilities (*Banker et al., 2020*). Apraxia of speech ultimately impacts the ability to produce precise articulation (*Allison et al., 2020*). The recognition and mitigation of these impairments are crucial to foster inclusive education and provide personalized assistance to students who are impacted by them. To promote inclusive education and provide adequate assistance to students with learning disabilities, educators and support professionals need to acknowledge and effectively address these concerns using the latest technologies (*Yenduri et al., 2023a*).

## Inclusive higher education

The concept of inclusion is subject to many interpretations among people. In higher education, the concept of inclusion refers to the ongoing and transformative efforts aimed at reorganizing higher educational institutions to effectively cater to the diverse needs of all students, with a special focus on those belonging to marginalized and disadvantaged groups (*Collins, Azmat & Rentschler, 2019*; *Li, Li & Luo, 2021*). Ensuring the inclusion of diverse learners and addressing their unique needs is of paramount importance, despite the many interpretations and definitions of inclusion that may exist. When considering inclusivity

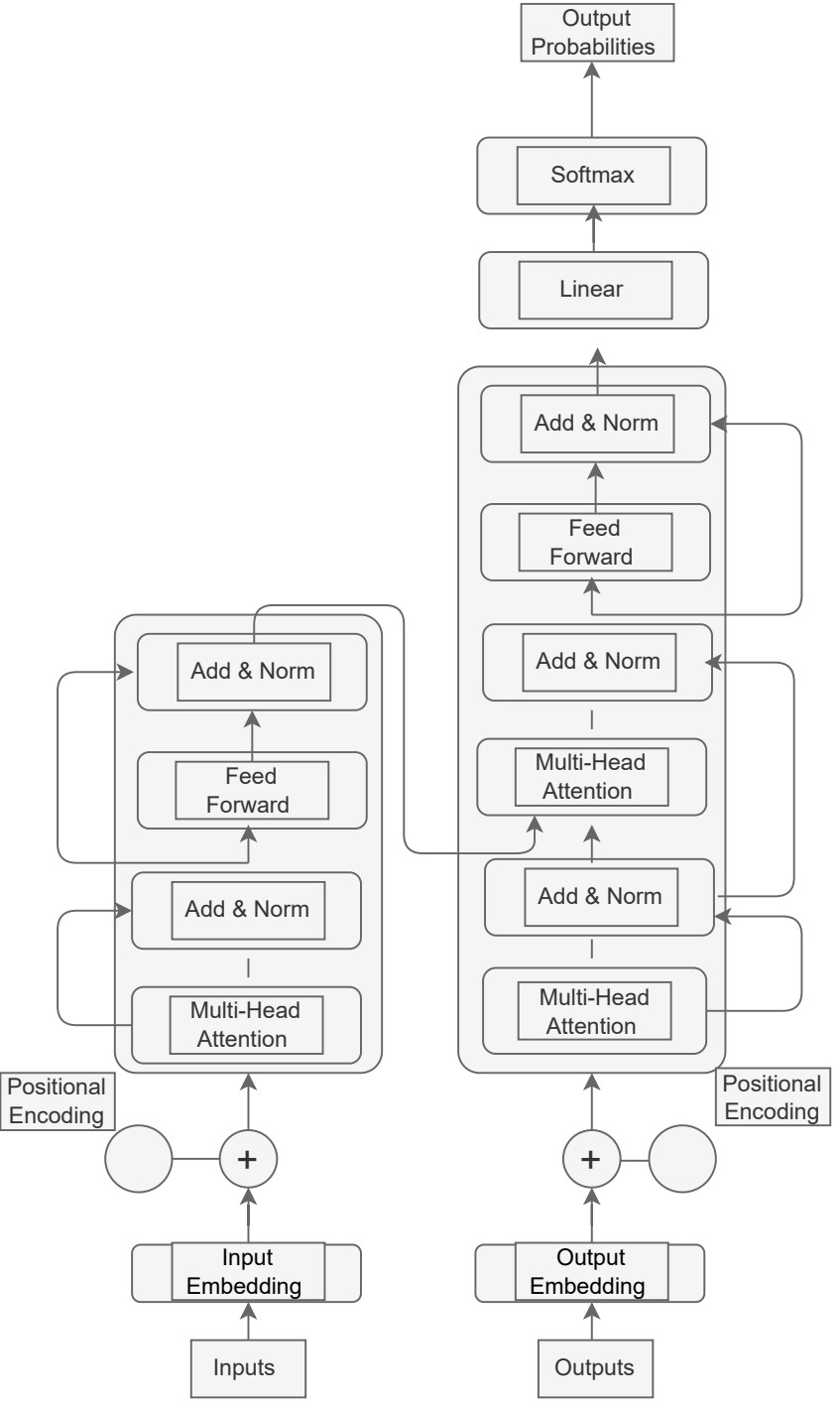

**Figure 2    General transformer architecture of GPT.**

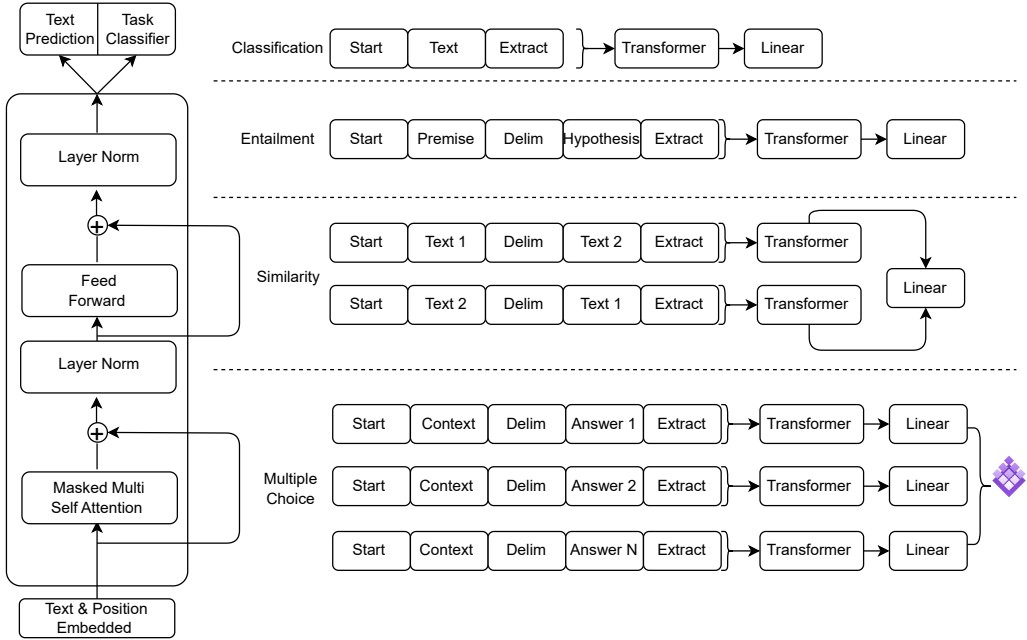

**Figure 3** **Transformer architecture and input transformations for fine-tuning on different tasks.**

within an institution, it is important to adopt a broad viewpoint that encompasses all people. Achieving inclusivity within a higher education system necessitates not only the establishment of requisite structures, such as inclusive curriculum, accessible architecture, and enhanced accessibility, but also actively combating the maintenance of inequality (*Arday, Zoe Belluigi & Thomas, 2021*). The objective is to advocate for social justice in education, particularly for historically marginalized students, and to promote a transparent and equal educational system.

## Motivation and contributions

The need to increase learning accessibility, particularly for those with learning disabilities, with the help of modern technology is what motivates us to conduct a review of the role of GPT in promoting inclusive education for people with learning disabilities. The use of AI conversation models, such as GPT-3.5 and GPT-4, in institutions of higher education that aspire to create an inclusive learning environment that accommodates the unique requirements of each student, regardless of their learning disabilities. GPT can provide students with personalized assistance, facilitate interactive and adaptable learning experiences, and provide timely feedback (*Ilieva et al., 2023*; *Alyasiri & Ali, 2023*). By the fundamental principles of inclusive education, GPT can adapt to individual needs and offer multiple modes of interaction, such as vocal commands and text-based interfaces (*Mahmood et al., 2023*). GPT enables students with learning disabilities to successfully overcome obstacles, participate actively in the educational process, and acquire vital skills while ensuring that no one is excluded from the pursuit of knowledge. The audience for this review paper includes experts in natural language processing and generative artificial

intelligence and their applications in education. Principal contributions of this review include:

- a complete overview of GPT, learning disabilities, and inclusive higher education is presented;
- the impact of GPT on promoting inclusive higher education for people with various learning disabilities is also highlighted;
- some of the most important research and industry projects related to promoting inclusive higher education for people with various learning disabilities are discussed;
- several challenges associated with the incorporation of GPT in promoting inclusive higher education for people with various learning disabilities are discussed. In addition, we also provided possibilities for future research that encourage researchers and industry to pursue further investigation.

## IMPACT OF GPT IN INCLUSIVE HIGHER EDUCATION

GPT provides tailored assistance for students with learning disabilities using the design of personalized learning materials, interactive resources, and easily accessible information. The platform has the ability to accommodate different learning styles by offering text-to-speech capabilities for those with learning disabilities and generating interactive activities for students who prefer more dynamic learning approaches. GPT also provides immediate feedback, enabling tailored strategies for learning and successfully meeting various educational requirements. This technology enables a more comprehensive learning environment, addressing the various difficulties encountered by each learner. This review explores the many uses of GPT in offering linguistic assistance, creating inclusive educational resources, _etc._ In addition, the review will also discuss the challenges and ethical concerns associated with using GPT to promote inclusive higher education. The following section provides a detailed overview of the impact of GPT in promoting inclusive higher education for students with various learning disabilities, as depicted in Fig. 4. It is noteworthy that GPT has thus far achieved a level of capability that enables it to effectively process and analyze text and images. However, when it comes to audio or video, GPT necessitates the support of third-party software applications.

### Personalized learning

Personalized learning is a method of customizing lessons for each individual based on their background knowledge, learning style, needs, and interests (_Basham et al., 2016_). People with a wide range of learning disabilities can benefit greatly from the personalized learning that GPT can provide. GPT, by understanding various learning profiles, effectively engages individuals by considering their unique strengths and preferred styles and overcoming obstacles imposed by their learning disabilities. GPT allows individuals to learn at their own pace, promotes equality by adapting to needs, and boosts confidence through individualized successes. This helps improve their abilities by focusing on their strengths, training them with lifelong learning skills, and empowering them to take care of their education and excel in academics.

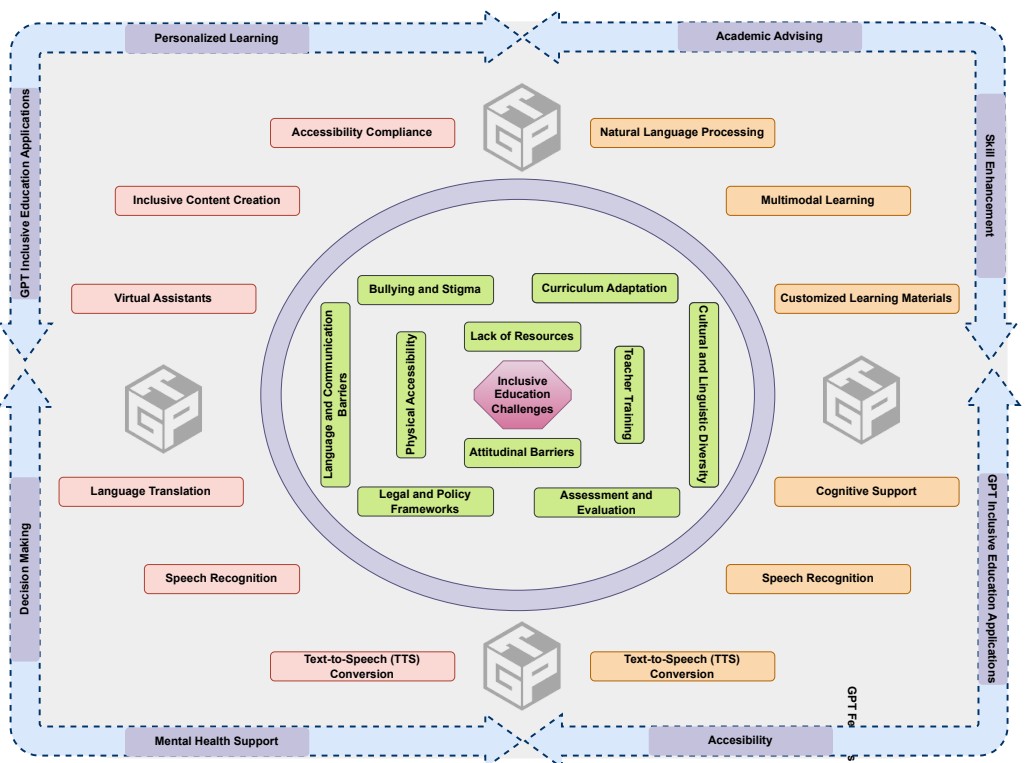

**Figure 4** **Impact of GPT in inclusive higher education.**

GPT can also create customized learning materials based on the specific needs of students with learning disabilities (*Floridi & Chiriatti, 2020*). For instance, GPT can generate simplified explanations, provide visual help, or provide audio content based on the student's preferences and learning style. As a result, complex subjects become easier to understand and more engaging. In addition to that, GPT can offer learning materials in different customized formats, such as braille, large print, or audio. GPT can swiftly convert standard materials into any of these formats, promoting that all students have equal access to information. The analyzing ability of GPT can also help in personalizing the teaching approach based on a student's progress and responses (*Lund & Wang, 2023*; *Zhao et al., 2022*). GPT can identify key areas of difficulty in understanding and can generate additional explanations based on the requirements. This dynamic adaptation can help students overcome challenges more effectively.

Additionally, GPT can provide instant feedback on assignments and assessments (*Liu et al., 2021*). This helps the students with learning disabilities identify their strengths and areas that demand improvement. This instant feedback provided by GPT can significantly benefit students with learning disabilities who may require extra care. Some learning disabilities that affect communication skills, such as speech disorders, language disorders, and autism spectrum disorder (ASD), can also be assisted by GPT. It can serve as a communication tool, helping students express their thoughts and ideas more effectively (*Kasneci et al., 2023*).

It can communicate information either through text, or by connecting to a third-party speech-generation application.

GPT can assist in creating a supportive learning environment for students with learning disabilities (*Yin et al., 2022*). As a result, adopting GPT in inclusive higher education will improve the independence of people with learning disabilities, and make them ready to compete with their peers. In addition, GPT in the future can also be seamlessly integrated with most existing assistive technologies. This can enhance their abilities and make them even more effective tools for inclusive education. Therefore, through personalized learning, GPT promotes inclusive higher education for people with various learning disabilities.

## Academic advising

Academic advising is the process through which a student and an academic adviser discuss educational and professional goals, examine the benefits of general education, examine the institution's services and regulations, and choose the right courses (*Young-Jones et al., 2013*). GPT can analyze a student's learning profile and patterns, including their strengths, weaknesses, and learning styles. Based on the analysis, it can then recommend courses that align with their interests and accommodate their learning disabilities (*Dale, 2021*). For instance, assuming that a student has dyslexia, GPT would recommend courses with more visual content or audio lectures, which can help students access content effectively and on par with their peers.

GPT can communicate with students using various forms to accommodate different needs. For students with reading difficulties, it can generate audio explanations or visual diagrams to deliver information effectively (*Bubeck et al., 2023*). This confirms that communication is accessible to all students, regardless of their learning disabilities. GPT can also help in understanding complex topics by giving simplified explanations. For instance, when students struggle to understand complex academic curricula due to their learning disabilities, GPT can simplify it by breaking it down into simpler terms. It can provide step-by-step explanations and analogies by considering the student's cognitive abilities (*Lee, Bubeck & Petro, 2023*). This can enhance their understanding and learning experience.

GPT can also assist in monitoring academic progress. For instance, students with learning disabilities may find academics challenging. GPT can offer clear and brief guidance on inferring requirements, generating outlines, and offering tips to effectively tackle these academic challanges. It can also aid in preparing for their future (*Elkins & Chun, 2020*).

GPT can create customized academic questions based on students' learning styles and disability-related challenges. It can provide feedback and strategies for effective academic management, ways for effective time management during academic activities, and reduce anxiety for students with learning disabilities (*Zaib, Sheng & Emma Zhang, 2020*; *Huang et al., 2023*).

GPT can also help in providing continuous academic feedback for the efforts made by students with disabilities. It can analyze progress and provide personalized academic assistance. For instance, if a student has any difficulty organizing their thoughts on academics due to a learning disability, GPT can help by offering guidance and enhancing

their thinking. Therefore, through academic advising, GPT promotes inclusive higher education for people with various learning disabilities.

## Accessibility

Accessibility is the process of making information, activities, and/or environments sensible, meaningful, and usable for the widest possible audience (*Bright, 2022*). In higher education, accessibility for people with learning disabilities is crucial for building an inclusive and equitable society (*Nori et al., 2023*). Learning disabilities comprise a range of neurological conditions that can affect an individual's ability to acquire, process, or express information effectively. Assuring accessibility in various aspects of life, including education, technology, and communication is vital to providing equal opportunities and simplifying meaningful participation for everyone, irrespective of their cognitive differences (*Zhang et al., 2023b*). By prioritizing accessibility, we encourage the development of an environment where individuals with learning disabilities can fully employ their skills, learn, and contribute. This ultimately promotes their personal growth, social integration, and overall well-being.

GPT cannot be directly used to convert text to speech or speech to text. However, GPT can be deployed into applications that offer these functions through additional add-on technology layers (*Wolf et al., 2020*; *Dida, Chakravarthy & Rabbi, 2023*). The GPT processes the provided input text to produce a coherent and contextually appropriate spoken response. This GPT-generated text output can be sent to a speech synthesis engine. This engine converts the text into natural-sounding speech. Likewise, the vice versa is also possible with GPT (*Quintans-Júnior et al., 2023*). GPT's ability to convert written text into spoken language and vice versa could be a game-changer for people with various learning disabilities. Text-to-speech conversion possibilities allow individuals with dyslexia or visual impairments to use educational materials in auditory form (*Hu et al., 2020*). This can help people with disabilities reduce the obstruction of struggling with reading and allow them to focus on understanding the content. On the other hand, speech-to-text conversion enables students who have difficulty expressing their thoughts through writing to participate enthusiastically in discussions, assignments, and assessments.

Visual content, such as images, graphs, and charts, has become an integral part of education. However, it can pose difficulties for students with visual impairments (*Zhang et al., 2023c*). GPT can also generate alternative text descriptions for visual elements, making them accessible to screen readers. GPT-4 can read and analyze the images. Through its ability to analyze the image, it can gain an understanding of the visual content. Using its language-generation abilities, it can generate concise and informative text that describes the content of the image.

Students who do not possess bilingual talents, in addition to having learning disabilities, may have difficulty understanding academic content presented in a language other than their native tongue. GPT has the potential to address this disparity by facilitating the translation of educational materials into a language that aligns with the student's linguistic proficiency and comfort level. GPT can be trained in the language translation process (*Shafeeg et al., 2023*). It can be given with the input in a source language. Using its analytical ability, it understands the context, meaning, and nuances of the given text in the source

language (*Hämäläinen, Tavast & Kunnari, 2023*). Using its language generation ability, it can generate coherent and contextually accurate translations of the input text in the desired target language. This confirms that language obstacles do not affect the student's learning experience, making education more accessible and inclusive (*Othman & Al Mutawaa, 2023*).

GPT can also help students with communication disorders. They may find it difficult to express thoughts, ideas, and questions. For instance, students with communication disorders provide input through various sources, such as typing, selecting options, or using a communication device (*Balkus & Yan, 2022*). GPT can understand the context of the user's input and can generate an apparent response message to the user's query (*Ressmeyer, Masling & Liao, 2019*; *Sakirin & Said, 2023*). GPT-enabled communication tools can generate text or speech based on the input provided by the student. This can assist them in expressing themselves more clearly and effectively, making them better communicators with peers and educators. Therefore, through providing accessibility, GPT promotes inclusive higher education for people with various learning disabilities.

## Mental health support

Mental health support encompasses the provision of assistance aimed at safeguarding or enhancing individuals' mental health and psychosocial well-being (*Fukuti et al., 2020*; *Catherine, Towfek & Abdelhamid, 2023*). GPT can analyze and understand the emotions conveyed by students through their writing or speaking. GPT, with its adaptive language processing skills, can adapt to the language patterns of individuals with disabilities. This includes their unique style of writing or speaking (*Mathew, 2023*). It can also understand emotional cues in the input. This is particularly important for the students struggling to convey emotions verbally or through any traditional means. It can also generate empathetic responses that validate the emotions expressed. With these abilities, GPT can recognize signs of distress, anxiety, or other emotional challenges in students with learning disabilities and act accordingly.

Students with disabilities must expect to share their thoughts and problems with someone. A non-judgmental AI system like GPT can be an appropriate replacement for a human being (*Reed, 2021*). They might find it easier to communicate their concerns to GPT. It can offer a safe and confidential space for students to convey their emotions and feelings without fear of stigma.

GPT can be trained on users' preferences based on the inputs given. With this ability, it can understand their interests, challenges, and needs. This understanding can help the GPT analyze their preferences, challenges, and requirements (*Trajtenberg, 2018*). Based on this, GPT can suggest appropriate resources, such as articles, videos, websites, coping strategies, relaxation methods, and self-care tips based on the individual's expressed needs. It can also be used to summarise the content suggested. This can help students get an overview of the material in a short time. This support provided by GPT can empower students to manage their mental health effectively.

GPT can simplify the process of collaborating with others for people with learning disabilities by being a supportive intermediary tool, helping communication, generating

ideas, and assisting in various collaborative activities. GPT can support individuals with learning disabilities in expressing their thoughts and ideas more effectively (_Beerbaum, 2023_). It can help generate ideas for group projects, collaborative brainstorming sessions, or creative endeavors. It can provide a wide range of recommendations based on the input it obtains from collaborators. It can also summarise lengthy discussions or meetings, confirming that key points are captured for future reference. This helps individual students with learning disabilities who are facing challenges processing large amounts of information. It can also be used to assist in clarifying complex concepts. It can provide simplified explanations, analogies, or visual representations that improve understanding for individuals with learning disabilities (_Katz et al., 2023_). It can also offer constructive feedback on written content or project drafts by the students. This feedback can be specifically useful for individuals with learning disabilities, who may need additional help in reviewing and revising their work. GPT can also support collaborative writing projects by suggesting sentence structures, grammar improvements, and vocabulary improvements. Overall, GPT, with its abilities, can aid students with learning disabilities to compete with their peers while maintaining good mental health.

## Decision-making and skill enhancement

The act of decision-making involves the identification of a particular decision, the collection of relevant information, and the evaluation of several possible options. Decision-making is a necessary skill that allows individuals to shape their lives and navigate a complex world. For students with learning disabilities, the significance of effective decision-making takes on a heightened impact. These individuals frequently encounter unique challenges that can cause difficulties in the decision-making process. Learning disabilities in a student may impact information processing, comprehension, and communication (_TS2 Space, 2023_). It can make it more difficult to gather, analyze, and synthesize the information needed to make informed choices. Additionally, decision-making includes cognitive intelligence, which can be challenging for students with learning disabilities. These challenges can lead to feelings of uncertainty, anxiety, and a fear of making the wrong choice. Therefore, making use of decision-making skills customized to the needs of students with learning disabilities is critical. Providing these students with appropriate decision-making tools can help them overcome challenges and gain confidence to navigate through the opportunities and choices that lie ahead.

GPT serves as an instrumental tool for individuals with learning disabilities in higher education by offering decision-making support. It can examine a student's academic history, learning choices, and disability-related challenges to provide customized recommendations. For instance, it can suggest precise courses that align with a student's strengths and interests. In addition, it can also recommend study strategies that cater to their specific learning style (_Bundhoo, 2023_). This guidance allows students to make informed decisions about their educational path. It also ensures that they choose courses and resources that maximize their chances of success. For instance, imagine a student with dyslexia enrolling in higher education. The student may not be sure about which course to choose, fearing that heavy reading necessities may hinder their progress. GPT can be trained on this by analyzing

their academic background and learning choices. Based on the data provided, GPT can recommend courses with minimal reading assignments, such as math or computer science, where visual and hands-on learning are essential. This customized guidance allows the student to select a course that aligns with their strengths and simplifies their learning challenges.

GPT can also act as a personalized tutor, customizing its approach to meet the unique learning needs of students with disabilities. It can offer real-time explanations of complex concepts, provide practice exercises for better understanding, and adapt content based on the student's progress. For instance, if a student struggles with mathematical concepts, GPT can offer step-by-step explanations and additional practice problems, gradually building the student's proficiency. This adaptability improves skill development and concept comprehension, ultimately helping to boost academic performance. For instance, imagine a college student with attention-deficit/hyperactivity disorder (ADHD) who struggles to maintain focus during lecture hours. Assume that this student has enrolled in a challenging physics course, and the complex concepts and lengthy lecture sessions often overwhelm them. To address this challenge, GPT can provide personalized support (*Zaremba & Demir, 2023*). During each lecture, it can be trained to generate real-time summaries and simplifications of the key concepts discussed. This can break the complex challenge down into manageable chunks. In addition, GPT can be trained to offer interactive practice problems related to the lecture content with instant feedback. As a result, the student's understanding of physics concepts improves, and their confidence in the subject grows. This instance illustrates how GPT can improve skills and concepts for students with learning disabilities by providing customized explanations and practice materials aligned with their specific requirements. As the student gradually adjusts to the regular academic tasks, their skills and self-assurance increase, enabling them to make more independent decisions.

In summary, we recommend that educators integrate GPT into their teaching methods for personalized content delivery, while policymakers should advocate for and fund GPT-based educational initiatives (*Liu, Subbareddy & Raghavendra, 2022*). Higher education institutes are advised to adopt GPT tools for inclusive curriculum design, ensuring accessibility for all learning abilities. Educational technology developers should focus on creating GPT applications tailored to diverse learning disabilities, emphasizing user-friendly interfaces and customizable features (*Attili et al., 2020*). Collaboration among these stakeholders is essential for maximizing GPT's potential in fostering an inclusive higher educational environment.

## PROJECTS

This section outlines the related projects that have been built with GPT approaches to promote inclusive education for people with learning disabilities.

### Be My Eyes

Be My Eyes is making great strides in supporting inclusive education for those with learning difficulties in partnership with OpenAI's GPT-4. The launch of the GPT-4-powered Virtual Volunteer tool represents a significant advancement in terms of information accessibility,

usefulness, and availability for those who are blind or have limited vision. Contextual understanding, feedback and support, independence and empowerment, accessibility in education, and personalized visual assistance are all provided by this project. The Be My Eyes Virtual Volunteer application is a prime example of how technology may be used to help people with learning difficulties overcome obstacles to education. Their provision of tailored, situation-specific, and instantaneous visual support is making a substantial contribution to the advancement of inclusive education and fostering a fairer learning environment for everyone (*Be my eyes, 2023*).

### Duolingo

One of the top suppliers of educational technology, Duolingo is well-known across the world for its creative use of applications to teach languages. Duolingo, which led the way in incorporating artificial intelligence into education, has now partnered with OpenAI to provide its users with English testing, automated feedback, and personalized learning. This is made possible by the use of GPT-4, the company's most sophisticated language model to date. Having more than 500 million users globally, Duolingo aims to be the best online language teacher by providing each student with individualized attention. Interestingly, every day, students complete a quarter of a billion activities on the site. By using this enormous amount of data and machine learning algorithms, Duolingo can better understand how individuals learn and improve its program to make language acquisition more effective. Due to this, Duolingo keeps changing the face of language learning, improving its effectiveness, accessibility, and level of engagement for students all around the world (*Duolingo, 2023*).

### Khanmigo AI

Khan Academy, a non-profit educational organization, is implementing a pilot initiative to provide Khanmigo AI to over 8,000 K–12 teachers and students. According to the project description provided by Khan Academy, Khanmigo is built around the underpinned version of the LLM of OpenAI's GPT-4, which has been trained using educational materials sourced from Khan Academy. To enhance the personalized and helpful response tone of Khanmigo, as well as to mitigate the speedy provision of responses to students, additional parameters have been included in the product (*Khan Academy, 2023*).

### Talk to Transformer

Talk to Transformer is an innovative chatbot for social-emotional learning that leverages GPT-3 to engage students in meaningful conversations about their feelings and emotions. This tool is useful for cultivating important social and emotional skills, such as communication, conflict resolution, empathy, self-awareness, and stress management. By generating customized responses and providing constructive feedback, Talk to Transformer assists students in the development of these vital skills, facilitating their success not only in academic settings but also in relationships and throughout life. It is a valuable and versatile resource for students of all ages, assisting them in navigating the complexities of emotions and interpersonal interactions (*Inferkit, 2023*).

### TutorEva

TutorEva is a GPT-3-based AI-powered assignment assistant program that provides invaluable assistance to students with learning disabilities. It provides personalized learning assistance, real-time feedback, and tools for enhancing study routines, all of which are designed to boost students' confidence and academic success. By providing individualized solutions and explanations, TutorEva accommodates a variety of learning disabilities, including dyslexia, dyscalculia, ADHD, and ASD, as evidenced by specific examples. TutorEva is a potent tool for parents and instructors of students with learning disabilities to enhance their students' comprehension, assignment completion, study skills, and overall academic confidence (*TutorEva, 2023*).

### Scaler

Scaler utilizes GPT-3′s capabilities to assist students with learning disabilities in multiple ways. It provides customized learning plans based on an individual's requirements and objectives, adaptive learning materials that adapt to a student's progress, real-time feedback for immediate corrections and progress monitoring, and social-emotional learning support. SCALER provides text-to-speech options, step-by-step solutions, task summaries, and opportunities for skill development for students with dyslexia, dyscalculia, attention deficit hyperactivity disorder, and autism spectrum disorder. In the end, the use of GPT-3 by SCALER enables students with learning disabilities to excel in both academic and life pursuits, thereby promoting a more inclusive and supportive educational environment (*Scaler, 2023*).

Based on the aforementioned projects, it is evident that GPT plays a significant role in advancing higher education opportunities for those with learning disabilities.

## CHALLENGES AND POTENTIAL SOLUTIONS

The following section explores the challenges and potential solutions of GPT in promoting inclusive higher education for students with various learning disabilities, as depicted in Fig. 5.

### User interface
#### *Challenge*
Accessibility and effectiveness of the user interface (UI) of GPTs for students with learning disabilities remain significant issues within the context of inclusive education. While their classmates without learning disabilities can effortlessly do so for academic purposes, students with learning disabilities may struggle. Unfortunately, students with learning disabilities require a personalized GPT UI environment to facilitate their education. If not provided, the fundamental principle of inclusive education may be compromised as a result. The difficulty resides in designing GPT user interfaces that accommodate the specific learning patterns and requirements of students with learning disabilities. Consequently, the ongoing challenge is to design GPT user interfaces that are adaptable, accessible, and accommodating so that all students can completely participate and benefit from academic technology in a welcoming educational environment (*Bresnahan, 2023*).

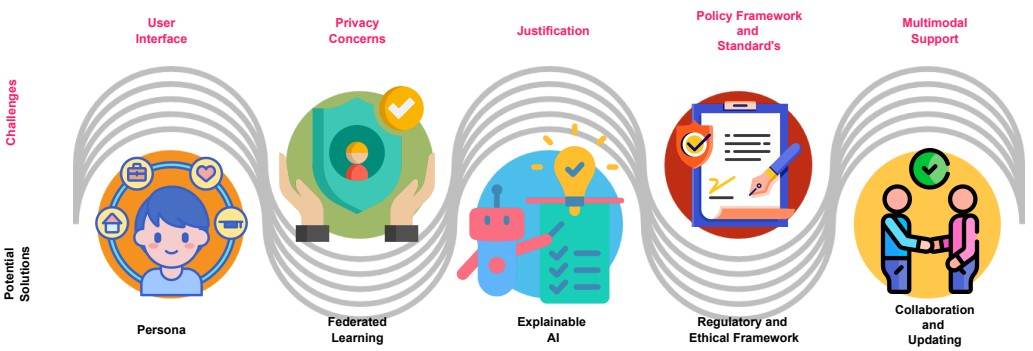

**Figure 5  Challenges and potential solutions.** Image sources: © Copyright 2024 Ofeex | PRESENTA-TIONGO; Safety free icon, Flat Icons; Persona free icon, Flat Icons; Explanation free icon, anilofex, Flaticon license; Insurance free icon, Chanut-is-Industries, Flaticon License; Partnership free icon, Freepik, Flaticon.

### Potential solution

Designing inclusive GPT UIs for people with learning disabilities in the context of inclusive education is a complex challenge. The use of personas in human–computer interaction (HCI) design is a potent solution to this problem. These personas represent diverse user profiles, including those with various cognitive difficulties, allowing designers to comprehend their unique requirements and obstacles (*Märtin, Bissinger & Asta, 2023*). By customizing user interfaces to meet these requirements, usability is enhanced and access barriers are reduced, validating UI enhancements through usability testing with users who match persona profiles. An iterative, empathy-driven approach to design, compliance with accessibility standards, and the development of guidelines all contribute to making GPT user interfaces more accessible and effective for students with learning disabilities. This strategy promotes inclusive education by ensuring that technology is beneficial for all students, regardless of their learning profiles.

## Privacy concerns
### Challenge

Developing a GPT that can provide personalized learning experiences to students with learning disabilities while maintaining the utmost data security and privacy standards is challenging (*Huallpa et al., 2023*). Students with disabilities may require interactions and materials that are tailored to their specific requirements and circumstances. Frequently, these interactions involve the disclosure of sensitive personal information. Achieving a balance between privacy and personalization is of the utmost importance. Concerns regarding data retention, storage, and third-party access in the use of GPT must be addressed to protect the privacy and personal information of students with learning disabilities.

### Potential solution

The implementation of GPT for students with learning disabilities, along with the adoption of federated learning, presents a robust solution to address concerns about privacy in

the context of personalized learning experiences. This approach only transfers model updates rather than raw data to centralized servers, guaranteeing data localization on students' devices and safeguarding the confidentiality of critical information. In addition to strengthening privacy, informed consent and data minimization practices give students the power to exercise decision-making in determining the extent of data they choose to share. By using federated learning, GPT can effectively support students with disabilities in their pursuit of education, while upholding strict data security and privacy protocols by achieving an appropriate balance between personalization and privacy (*Agrawal et al., 2022*).

## Justification
### *Challenge*

Providing an unbiased explanation of GPT results is challenging. In the context of GPT, this necessitates the provision of effective and justifiable recommendations and outcomes for students with disabilities. Due to their specific needs, disabled students frequently require individualised learning environments, highlighting the significance of the GPT's capacity to deliver precise and equitable outcomes. This work requires the elimination of biases, the formulation of recommendations that take into consideration the individual requirements and learning goals of each learner, and the provision of explicit justifications for the results. It is essential to establish protocols for detecting and eliminating biases within the GPT-generated recommendations to reduce the recurrence of erroneous assumptions and undesirable outcomes. Therefore, promoting inclusive education for students with disabilities through the use of GPT poses a significant challenge. This challenge involves the delicate task of striking a balance between ensuring equity and transparency in the explanations generated by the system and upholding the principle of customizing each individual's educational experience to their specific requirements (*Grassini, 2023*).

### *Potential solution*

The use of explainable artificial intelligence (XAI) within GPT presents a potential solution to the challenge of providing explanations for outcomes to those with learning disabilities. XAI ensures transparent decision-making through various strategies, including the mitigation of biases, customization of justifications for individual learning disabilities, provision of a user-friendly interface for easy access to explanations, establishment of a feedback loop for enhanced interaction, and provision of clear explanations for GPT-generated recommendations. The integration of XAI with GPT has the potential to enhance trust, fairness, and transparency in learning results. This coincides with the principles of inclusive education and ensures that students with disabilities get fair and understandable support (*Susnjak, 2023*).

## Policy framework and standards
### *Challenge*

Incorporating GPT into inclusive higher education for students with learning disabilities can cause appropriate concerns regarding potential legal consequences. In the educational system, the lack of a unified policy framework and standardized regulations on the use of

GPT is a significant obstacle. In the absence of appropriate norms, potential issues arising from the use of GPT to assist students with learning disabilities have the potential to not only lose trust, but also reduce the GPT's overall effectiveness in meeting the unique needs of these students (*Chan, 2023b*).

### Potential solution

In order to address the challenge-related policy framework and standards for using GPT, it is critical to establish a comprehensive regulatory and ethical framework to effectively address these issues and provide a safe, equitable, and legally binding GPT-assisted educational environment for students of all abilities. This framework serves the dual purpose of protecting the rights of students and promoting the appropriate and beneficial use of GPT within the context of inclusive education.

## Multimodal support
### Challenge

The absence of multimodal support in GPT models is a significant challenge in promoting inclusive higher education for individuals with learning disabilities. These models cannot provide a holistic learning experience if they cannot simultaneously process and generate multiple modalities, such as text, illustrations, and audio. It is difficult for students with learning disabilities to completely interact with educational materials due to this lack of accessible information that is adapted to individual learning styles and preferences. For all students, regardless of learning disabilities, to benefit equally from higher education, GPT models must incorporate multimodal capabilities as a bridge.

### Potential solution

To address the problem of multimodal support in GPT models for promoting inclusive higher education for students with learning disabilities, some ideas include creating dedicated multimodal models, keeping training materials up to date, adding assistive technologies, committing to ethical data management practices, making user-friendly interfaces, improving existing models all the time, and starting collaborations with educators and researchers. These measures can help in the creation of an accessible, adaptable, and effective multimodal GPT.

## CONCLUSION AND FUTURE DIRECTIONS

The primary goal of this review is to understand the role of GPT in promoting inclusive education for students with learning disabilities in the context of higher education. During the course of this investigation, a comprehensive exploration of several online digital libraries was conducted to identify scholarly journal articles and conference papers that are relevant to our research. Based on our comprehensive review of the relevant literature, it is evident that there is a lack of reviews related to GPT in promoting inclusive education for students with learning disabilities in the context of higher education. Our analysis has further identified projects related to the use of GPT in the context of inclusive education and supporting students with learning disabilities in higher education. Furthermore, we have highlighted the challenges associated with using GPT in inclusive education for higher

education students with learning disabilities. Additionally, we presented various potential solutions to address these challenges. In conclusion, we provided a comprehensive review for establishing a higher education system that is both accessible and inclusive, with a particular focus on using GPT to facilitate the educational advancement of students with disabilities. Potential future directions include improving GPT's adaptability to accommodate a wide range of learning styles and integrating it more extensively into educational platforms to provide immediate assistance (*Chen et al., 2023*). In addition, continuous research should focus on enhancing GPT's comprehension of complex learning challenges in order to provide equal access to education for all learners.

### Funding

The authors received funding in the form of a grant from the project "DIVERSASIA –Embracing Diversity in Asia through the Adoption of Inclusive Open Practices" with Project No: 618615-EPP-1-2020-1-UK-EPPKA2-CBHE-JP by Erasmus + Programme of the European Union to perform this work. The authors also received financial support from the Vellore Institute of Technology. The funders had no role in study design, data collection and analysis, decision to publish, or preparation of the manuscript.

### Grant Disclosures

The following grant information was disclosed by the authors:
The project "DIVERSASIA –Embracing Diversity in Asia through the Adoption of Inclusive Open Practices" by Erasmus + Programme of the European Union: 618615-EPP-1-2020-1-UK-EPPKA2-CBHE-JP.
The Vellore Institute of Technology.

### Competing Interests

Wei Wang is an Academic Editor for PeerJ Computer Science. Thippa Reddy Gadekallu is employed by Zhongda Group part-time as chief scientist.

### Author Contributions

- Thippa Reddy Gadekallu conceived and designed the experiments, prepared figures and/or tables, authored or reviewed drafts of the article, and approved the final draft.
- Gokul Yenduri performed the experiments, prepared figures and/or tables, and approved the final draft.
- Rajesh Kaluri performed the experiments, prepared figures and/or tables, and approved the final draft.
- Dharmendra Singh Rajput analyzed the data, prepared figures and/or tables, and approved the final draft.
- Kuruva Lakshmanna analyzed the data, prepared figures and/or tables, and approved the final draft.
- Kai Fang analyzed the data, authored or reviewed drafts of the article, and approved the final draft.

- Junxin Chen analyzed the data, authored or reviewed drafts of the article, and approved the final draft.
- Wei Wang conceived and designed the experiments, authored or reviewed drafts of the article, and approved the final draft.

## Data Availability

This is a literature review.

## Supplemental Information

Supplemental information for this article can be found online at http://dx.doi.org/10.7717/peerj-cs.2400#supplemental-information.

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
