# Peer review of "The role of GPT in promoting inclusive higher education for people with various learning disabilities: a review"

_PeerJ Computer Science, doi:10.7717/peerj-cs.2400_

## Round 0.1 · original submission · Major Revisions

The reviewers agree that the work is interesting. However, some issues need to be addressed. Please revise the work accordingly. Then is will be evaluated again.

Reviewer 1 ·

Basic reporting

The authors have conducted a comprehensive review on using the Generative Pre-trained Transformer (GPT) in the field of higher education. The possible challenges and constraints associated with the integration of GPT into inclusive higher education along with the potential solution. This review paper is interesting however there are some drawbacks that the authors should address them to improve this review paper.

Experimental design

This study must re-design the structure of this review.

Validity of the findings

Not available.

Additional comments

The authors have conducted a comprehensive review on using the Generative Pre-trained Transformer (GPT) in the field of higher education. The possible challenges and constraints associated with the integration of GPT into inclusive higher education along with the potential solution. This review paper is interesting however there are some drawbacks that the authors should address them to improve this review paper.
1. The novelty points of this review paper must be indicated clearly.
2. In table 1, the specific field of GPT application must be indicated clearly corresponding to its contribution.
3. The development of GPT should be declared to lead to this review paper.
4. In figure 1, the authors provided the organization of this review. However, the interconnection between each part must be indicated and analyzed.
5. The relationship between each investigated part must be indicated clearly.
6. The contribution of each part in this review must be provided clearly.
7. The future direction of GPT should be indicated and analyzed as well as its significance.

Reviewer 2 ·

Basic reporting

.

Experimental design

.

Validity of the findings

.

Additional comments

• What specific features of GPT make it particularly effective in improving inclusivity and accessibility for students with learning disabilities in higher education?
• Elaboration on the personalized and diverse solutions that GPT can provide for students with learning disabilities, and how these solutions address their distinct requirements can be included.
• Based on which methodology or criteria employed in your review to assess the effectiveness of GPT in enhancing accessibility and inclusivity in higher education?
• More insights into the challenges and constraints associated with the integration of GPT into inclusive higher education, and what potential solutions were identified during your research should be discussed.
• What recommendations does your review offer for educators, policymakers, higher education institutes, and educational technology developers to effectively make use of GPT in promoting inclusive higher education for people with various learning disabilities? This section should be the key focus of this work.

• Related work that could be cited: < Liu, L., Subbareddy, R., & Raghavendra, C. G. (2022). AI Intelligence Chatbot to Improve Students Learning in the Higher Education Platform. Journal of Interconnection Networks, 22(Supp02), 2143032.> and < Attili, V. R., Annaluri, S. R., Gali, S. R., & Somula, R. (2020). Behaviour and emotions of working professionals towards online learning systems: sentiment analysis. International Journal of Gaming and Computer-Mediated Simulations (IJGCMS), 12(2), 26-43.>

·

Basic reporting

• The abstract provides a good overview of the article's objectives and scope, but it could benefit from a more succinct summary emphasizing key findings and contributions. Consider revising to enhance clarity and focus.
• The introduction effectively sets the context but lacks a clear thesis statement or specific objectives. Clarifying these early in the paper would provide a more directed narrative.
• The literature review is thorough, but there's a noticeable gap in directly linking the discussed studies to the specific context of higher education and learning disabilities. Integrating these links would strengthen the relevance of the review.
• The organization of the paper is logical, yet the transitions between sections are abrupt. Smooth transitions would enhance the flow and readability.

Experimental design

• The methodology section is well-detailed but lacks a critical discussion on the selection criteria's limitations and potential biases. Addressing these aspects would enhance the credibility of the review process.
• The inclusion and exclusion criteria are well-defined; however, the rationale behind these criteria choices is not sufficiently explained. Expanding on why certain criteria were chosen would add depth to the methodology.
• There's an absence of a clear framework or theoretical model guiding the review. Incorporating such a framework would strengthen the methodological rigor.

Validity of the findings

• The discussion on the impact of GPT in inclusive higher education is insightful but occasionally speculative. Grounding these discussions more firmly in existing literature or empirical data would improve the validity.
• The paper makes several assertions about GPT’s potential without adequately addressing the limitations or challenges of GPT itself. A balanced view including potential drawbacks would provide a more nuanced perspective.
• The conclusions drawn are relevant but lack a strong connection to empirical evidence or specific examples from the reviewed literature. Strengthening these connections would enhance the persuasiveness of the conclusions.

Additional comments

• The paper is a valuable contribution to the field, especially in highlighting the potential of GPT in inclusive education. However, exploring more diverse perspectives, including critical viewpoints, would enrich the discussion.
• The review covers a broad range of applications of GPT, but a more focused examination on specific aspects relevant to learning disabilities in higher education would make the paper more impactful.
• The future research directions are promising, but they could be more specific and actionable. Providing clear guidelines or specific research questions would be beneficial.

---

## Round 0.2 · accepted · Accept

Thanks to the authors for their efforts to improve the work. This version successfully satisfied the reviewers. It can be accepted. Congrats!

Reviewer 1 ·

Basic reporting

This review paper is well prepared now.

Experimental design

This study design is fine.

Validity of the findings

No comment.

·

Basic reporting

The paper is generally well-structured and written in clear English. Adequate background and context are provided on GPT models and inclusive higher education. The literature review appears comprehensive, covering relevant works on AI in education for students with learning disabilities. Figures like Fig. 1 and Fig. 4 help illustrate key concepts.

Experimental design

The review methodology is appropriate, covering a broad range of sources including academic databases and industry reports. The paper logically organizes content into sections on GPT basics, impacts on inclusive education, projects, and challenges. The coverage of topics like personalized learning, accessibility, and mental health support demonstrates a thorough exploration of GPT applications for students with disabilities.

Validity of the findings

The conclusions summarize key points on GPT's potential in inclusive higher education, while noting challenges like privacy concerns and need for multimodal support. Future research directions are identified, such as improving GPT's adaptability for diverse learning styles. The discussion of ethical considerations and policy implications adds depth to the analysis.

Additional comments

While the revision may not have fully addressed all previous review comments, the overall quality and comprehensiveness of the paper appear to be at an acceptable level for publication. The thorough coverage of GPT applications, challenges, and future directions in inclusive higher education provides a valuable contribution to the field.